# Mode Conversion of the Edge Modes in the Graphene Double-Ribbon Bend

**DOI:** 10.3390/ma12234008

**Published:** 2019-12-03

**Authors:** Lanlan Zhang, Binghan Xue, Yueke Wang

**Affiliations:** 1Department Medical Technology and Engineering, Henan University of Science and Technology, Luoyang 471000, China; zhanglan80515@163.com; 2Jiangsu Provincial Research Center of Light Industrial Optoelectronic Engineering and Technology, Jiangnan University, Wuxi 214122, China; zpsu622@163.com

**Keywords:** surface plasmon polaritons, edge mode, graphene, mode conversion

## Abstract

In this paper, a new kind of graphene double-ribbon bend structure, which can support two edge graphene surface plasmons (EGSPs) modes, is proposed. In this double-ribbon bend, one edge mode can be partly converted into another one. We attribute the mode conversion mechanism to the interference between the two edge plasmonic modes. Based on the finite element method (FEM), we calculate the transmission and loss of EGSPs propagating along this graphene double-ribbon bend in the mid-infrared range under different parameters.

## 1. Introduction

Graphene, due to its unique mechanical, electrical and optical properties [1,2,3,4], is a promising candidate for nanoscale photonic applications in infrared frequencies. Surface plasmons (SPs) supported by graphene has recently attracted intensive attention driven by maturing state-of-the-art nanofabrication technology. Compared to SPs in noble metals, graphene surface plasmons (GSPs) exhibit even stronger mode confinement and relatively longer propagation distance, with an additional unique ability to be tunable by adjusting gate voltage or chemical doping concentration [5,6,7,8,9]. GSPs brings many unique phenomena such as negative refraction [10,11], cloaking [12,13], and superlens [14,15]. Its extraordinary features are applied to a series of optical devices such as absorbers [16,17], modulators [18,19,20], and sensors [21,22].

The graphene ribbons can support both waveguide GSPs (WGSPs) modes and strongly confined edge GSPs (EGSPs) modes. EGSPs are the fundamental modes, which are strongly localized along the graphene edge, show a larger effective refractive index and stronger field confinement [23]. EGSPs modes are firstly observed experimentally in a patterned graphene nanoribbon on Al_2_O_3_ substrates [24]. The width of the graphene ribbon is inversely proportional to the wave vectors of EGSPs and EGSPs modes show cut-off behavior [25]. Conventional straight ribbon waveguides, including multilayer nano-ribbon [26], ribbon resonators with rings [27] and wrings [28], are studied in detail. A graphene bending ribbon waveguide is proposed to explore the spatial coupling between the edge modes [29]. Furthermore, edge modes supported by bending ribbon waveguide spatially split with the strongly confined symmetric (anti-symmetric) mode, which shifts to the exterior (interior) edge of the incidence [30].

In the paper, we propose a new kind of double-ribbon bend that supports two EGSPs. Based on the finite element method (FEM), we firstly discuss the dispersion relation of EGSPs for this double-ribbon bend. We calculate transmission and loss of edge modes with different bending angles under Fermi level, separation distance, incidence wavelength, bending radius and double-ribbon width. The mode conversion is due to interference between the two EGSPs, and the loss of double-ribbon originates from the absorption loss and bending loss. The period of the mode conversion by simulation is in accordance with the theoretical results.

## 2. Structure and EGSPs Dispersion

The SPs supported by individual graphene nano-ribbons has been discussed very frequently. Plasmon interaction and hybridization in pairs of neighboring aligned ribbons are shown to be strong enough to produce dramatic modifications in the plasmon field profiles [31]. What will happen if there are two paralleled bending ribbons? Inspired by these edge modes study, we propose a new kind of bend structure with two paralleled graphene ribbons with an interval of *D*. When the two edges of two ribbons come closer to each other, they will bring out two of edge modes with opposite parity [31].

As illustrated in Figure 1a, a pair of paralleled graphene ribbon bends with a separation interval of *D* is deposited on the SiO_2_ substrate, and these two bending ribbons are of equal width. Figure 1b is the sectional view of Figure 1a. The dielectric above is air. The relative dielectric constant of SiO_2_ substrate is 2.25. The radius and the width of the double-ribbon bend are *R* and *W*, respectively. The thickness of the graphene double-ribbon bend is 1 nm. The bending angle of the structure is denoted by *θ*. Here, we only discuss the condition when *D* is smaller than 20 nm, where the ribbons structure supports the two EGSPs modes.

As is known, there is only one edge mode in a semi-infinite graphene sheet. When two semi-infinite paralleled graphene sheets are closed to each other, the two semi-infinite edge modes will be mutually coupled into two edge modes. Here, as width of the bending ribbon structure is much bigger than separation distance (*W* >> *D*), we can consider the two graphene ribbons as two semi-infinite sheets, and the coupling happens at the internal edge of the double-ribbon bend (near the blue region showing in Figure 1a). Here, the surface conductivity of graphene, *σ_G_*, is obtained using the Kubo formula [32]
(1)$$\begin{array}{l} \sigma_{G}=\frac{ie^{2}E_{f}}{\pi \hbar^{2}(\omega +i\tau^{-1})}+\frac{ie^{2}}{4\pi \hbar }In[\frac{2E_{f}-(\omega +i\tau^{-1})\hbar }{2E_{f}+(\omega +i\tau^{-1})\hbar }]+\frac{ie^{2}k_{B}T}{\pi \hbar^{2}(\omega +i\tau^{-1})}\mathrm{In}[\exp(-\frac{E_{f}}{k_{B}T})+1] \end{array}$$

In Equation (1), *ħ*, *e* and *k_B_* represent reduced Planck’s constant, the electron charge and Boltzmann’s constant respectively. Equation (1) shows that *σ_G_* depends on the Fermi levels *E_f_*, the momentum relaxation time *τ*, temperature *T*, and the photon frequency *ω*. Fermi energy level is shown in Equation (2)
(2)$$E_{f}=\hbar V_{f}{\left(\pi n\right)}^{1/2}$$
where *n* is the charge carrier concentration, the Fermi velocity *V_f_* is 10^6^ m/s. The permittivity *ε_G_* of graphene is governed by
(3)$$\varepsilon_{G}=1+\frac{i\sigma_{G}\eta_{0}}{k_{0}d}$$
where *η*_0_ (≈377 Ω) is the impedance of air, *τ* is chosen as 0.5 ps, the thickness of graphene *d* is 1 nm, and the temperature *T* = 300 K. Due to graphene’s particular characteristics, *ε_G_* can be tuned by modifying the gate voltage or doped by chemical doping.

We choose COMSOL Multiphysics based on the finite element method (FEM) to conduct modes analysis of GESPs mode for our double-ribbon bend and all simulation in the paper will be conducted by FEM [28]. Figure 1c,d plot GESP modes’ *z*-component of magnetic field in the *x*-*z* plane, when the incident wavelength is 6 μm, *E_f_* = 0.2 eV, *D* = 10 nm and *W* = 200 nm. *Z*-component of magnetic field in Figure 1c is symmetrical, which can illustrate the symmetric EGSPs mode (SEM); and that of Figure 1d is anti-symmetrical, which can illustrate the anti-symmetric EGSPs mode (AEM).

The differences between the real parts of effective refractive indexes for the two edge modes can be obtained by
(4)Δneff=Re(n1)−Re(n2)

In Equation (4), Re(*n*_1_) and Re(*n*_2_) represent the real part of two EGSPs’ effective refractive index respectively. Figure 2 shows the dispersion relation of the two EGSPs’ modes. It implies that Re (*n*_1_), Re (*n*_2_), and Δ*n_eff_* is related to the following parameters: Fermi levels *E_f_*, wavelengths *λ*, and separation distance *D*. Figure 2a shows the real parts of the effective refractive index (Re(*n_eff_*)) of the SEM (black line) and AEM (red line) mode both decrease with increasing *E_f_*; but the exact opposite is for the difference Δ*n_eff_* between two EGSPs’ modes (blue line), when *λ =* 6 μm and *D* = 10 nm. As can be seen in Figure 2b, when the incident wavelengths are 5, 6, 7, and 8 μm (plotted in pink, yellow, blue, and green lines) respectively, Δ*n_eff_* firstly rises then decreases with increasing Fermi level *E_f_* under the same wavelength. And the Δ*n_eff_* is larger when *λ* is increasing from 5 to 8 μm under the same *E_f_*. Figure 2c shows when separation distance *D* = 10, 12 and 14 nm (plotted in pink, yellow and blue lines), Δ*n_eff_* firstly rises then decreases with increasing Fermi level *E_f_* under the same separation distance *D*. The Δ*n_eff_* is larger when *D* is decreasing from 14 to 10 μm under the same *E_f_*. 

## 3. Mode Conversion and Simulations Results

Due to the interference between the SEM and AEM mode, mode conversion happens when one mode is propagating along with our proposed structure. We define a periodic variation angle *θ_T_*, which is used to analyze the interference process between the SEM and AEM [30,33,34]. The positive integer *k* represents the order of the conversion period
(5)$$2\theta_{T}\cdot R\cdot \Delta n_{eff}=k\lambda (k=1,2,3\ldots )$$

To further investigate the mode conversion mechanism between two EGSPs, we calculate bending loss, total losses and the mode distribution, under different bending angles *θ*, Fermi levels *E_f_*, separation distances *D*, wavelengths *λ*, and bending radii *R*, respectively. Here, the SEM is coupled into the bottom port of the double-ribbon bend as shown in Figure 1a, and propagates along the two ribbons structure. The initial parameters are: *D* = 10 nm, *R* = 300 nm, *λ* = 6 μm and *W* = 200 nm. We define the conversion efficiency *P*, which represents the ratio between the transmission of the converted mode and the total transmission at the output
(6)$$P=\frac{T_{2}}{T_{1}+T_{2}}$$
where *T*_1_ and *T*_2_ represent the transmission of SEM and AEM in respectively. *P* reaches the maximum, which means conversion efficiency between SEM and AEM is highest, and *P* reaches minimum, which means SEM propagates along our proposed structure without mode conversion.

Figure 3 shows the periodic conversion between two EGSPs’ modes under different Fermi levels (*E_f_* = 0.3, 0.4, 0.6, 0.8, and 1 eV). The SEM’s transmission is plotted in the black line and AEM’s is in the red line. The blue ones represent the conversion efficiency *P*, in Figure 3a–e. It is found that SEM is partly converted to AEM with different bending angles (ranging from 0° to 130°). The mode conversion happens due to the interference between the SEM and AEM. The three lines have the same change periods with a bending angle under the same *E_f_*. The variation amplitude of SEM’s declines with bending angle, but those of AEM and *P* are nearly unchanged. *θ_t_* is the conversion period, when the bending angle is *θ_t_**/*2, *P* reaches the maximum; and when the bending angle is *θ_t_*, *P* reaches the minimum. It is found that when *E_f_* is 0.3, 0.4, 0.6, 0.8, and 1 eV, *θ_t_* is 54.6°, 54°, 57.8°, 62.4°, and 67° respectively, which is accordance with Equation (5). The theoretical value *θ_T_* of the conversion period is 54.5°, 54.8°, 58.6°, 64°, and 69°, respectively. The maximum of conversion efficiency *P* is 15.40%, 13.92%, 11.86%, 8.68%, and 5.80%, respectively, and the conversion efficiency *P* decreases with increasing *E_f_*.

Because of graphene absorption loss and bending radiation loss, SEM’s transmission decreases gradually. Here, we define the total losses *L_T_*, which is calculated by
(7)$$L_{T}=1-T_{1}-T_{2}-R_{1}-R_{2}$$
where *R*_1_ and *R*_2_ are the reflectivity of SEM and AEM for our proposed structure. *L_T_* consists of absorption loss and bending loss. Here, we define this absorption loss of per unit length for a straight double-ribbon waveguide as normalized absorption loss *L*_a_. The normalized total loss *L*_t_ is derived from the total loss *L_T_* over the actual propagation length (i.e., the center arc length of this double-ribbon). The normalized bending loss *L*_b_ is defined by the difference between *L*_t_ and *L*_a_.

Figure 3f shows the total loss *L_T_* vs. bending angle under different fermi levels. *L_T_* increases with increasing bending angles under the same *E_f_*_,_ because larger bending angles mean larger propagation loss of EGSPs. *L_T_* decreases with increasing *E_f_* under the same bending angles. The three normalized losses *L*_t_, *L*_a_ and *L*_b_ are plotted by cyan, orange, and olive lines respectively in Figure 3g, they all decrease with increasing *E_f_*. Figure 3h shows the imaginary parts of the effective refractive index for SEM and AEM are almost the same, and Im(*n_eff_*) decreases with increasing *E_f_* for both SEM and AEM. Larger Im(*n_eff_*) causes a bigger normalized absorption loss *L*_a_, which can explain that *L*_a_ decreases with increasing *E_f_* in Figure 3h. When bending angels are 27.3° and 54.5°, the magnetic field distributions *H*_z_ are shown in Figure 3i,j under *E_f_* = 0.3 eV. It is found that when bending angle is 27.3°, SEM can be partly converted into AEM (*P* = 15.40%), so magnetic field distributions *H*_z_ at the output is the superposition of AEM and SEM; when bending angle is 54.5°, SEM can propagate through the bending structure without mode conversion, so magnetic field distributions *H*_z_ at the output is symmetrical.

The solid line and dashed line represent the theoretical value *θ_T_* and simulation value *θ_t_* for conversion period respectively, under different *E_f_*, in Figure 4. The conversion period firstly decreases and then increases with increasing *E_f_*_._ It is found that the theoretical value *θ_T_* matches very well with the simulation results.

Periodical mode conversion between SEM and AEM under different separation distances (*D* = 10, 12, and 14 nm) is shown in Figure 5a–c. Other parameters are as follows: *E_f_* = 0.6 eV, *R* = 300 nm, *λ* = 6 μm, and *W* = 200 nm. It is also found that SEM is partly converted to AEM varying with different bending angles. The transmission of two EGSPs and *P* both have the same changing period with bending angle (ranging from 0° to 90°) under the same *D* when *D* is 10, 12, and 14 nm, *θ_t_* is 58°, 65.7°, and 74°. The theoretical value *θ_T_*, which is obtained by Equation (5), is 59.2°, 68.0°, and 78.4°. *θ_t_* decreases with increasing *D*. That is because Δ*n_eff_* increases with increasing *D* under the same *E_f_* in Figure 2b. Meanwhile, the maximum conversion efficiency *P*_max_ respectively are 6.1%, 9.2%, and 13.5%, and increases with increasing *D*.

As illustrated in Figure 5d, the total loss *L_T_* is plotted in red, green, and blue lines when the separation distance *D* is chosen as 10, 12, and 14 nm. It is found that total loss *L_T_* increases with increasing bending angle under the same separation distance *D*, but *L_T_* is almost the same under the same bending angle. *L_T_* is a litter larger for smaller *D*, which is shown in the insert of Figure 5e. As shown in Figure 5e, the normalized loss *L*_t_, *L*_a,_ and *L*_b_ are plotted in cyan, orange, and olive lines, respectively. *L*_t_ and *L*_a_ decrease a little with increasing *D*. *L*_b_ barely changed with it. The reason is that Im(*n_eff_*) of SEM decreases slightly with increasing *D*, but the opposite is true for AEM, which is shown in Figure 5f. So the absorption loss of SEM decreases slightly with increasing *D*, but the opposite is true for AEM. The absorption loss mainly originates from SEM’s propagation loss. So when *D* is increasing, *L*_t_, *L*_a_ and total loss *L_T_* decreased a little due to the absorption loss difference between SEM and AEM. The proposed double-ribbon bend has the same curvature under three separation distances *D*, so *L*_b_ is also the same.

As shown in Figure 6a–d, periodic conversion between two EGSPs also happens under different wavelengths (*λ* = 5, 6, 7 and 8 μm). Other parameters are as follows: *E_f_* = 0.6 eV, *D* = 10 nm, *R* = 300 nm, and *W* = 200 nm. The transmission of two EGSPs and the conversion efficiency *P* both have the same change period with bending angles under the same wavelength. It is can be seen in Figure 6a–d that when the wavelength is *λ* = 5, 6, 7 and 8 μm, *θ_t_* is 54.9°, 57.8°, 63.5°, and 69.1°. Theoretical value *θ_T_* is 54.4°, 58.6°, 64.4° and 70.1° based on Equation (5). The maximum conversion efficiency *P*_max_ is 15.40%, 11.86%, 8.40%, and 4.96%, respectively. As illustrated in Figure 6e, the total loss *L_T_* is plotted in pink, yellow, blue, and green lines when the wavelength is 5, 6, 7, and 8 μm, respectively. It is found that total loss *L_T_* enhances with increasing bending angle under the same wavelength *λ* because of the absorption loss of EGSPs. *L_T_* decreases with increasing wavelength *λ* under the same bending angle. As shown in Figure 6f, the normalized loss *L*_t_, *L*_a_ and *L*_b_ are plotted in cyan, orange and olive lines, respectively. They all decrease with increasing *λ*. As shown in Figure 6g, Im(*n_eff_*) of SEM decreases with increasing *λ*, AEM does just the opposite. It’s worth noting that the proportion of SEM is much larger than AEM’s, the downtrend for Im(*n_eff_*) of SEM will play a dominating role in the absorption loss. Thus, *L*_a_ declines with increasing *λ*.

As shown in Figure 7a–d, periodical couplings between SEM and AEM are presented by four bending radii: 300, 400, 500, and 600 nm, respectively. Other parameters are as follows: *E_f_* = 0.6 eV, *D* = 10 nm, *λ* = 6 μm, and *W* = 200 nm. The transmission of two EGSPs and the conversion efficiency share the same converting period. When *R* is 300, 400, 500 and 600 nm, *θ_t_* is 57.8°, 43.3°, 35.0°, and 29.0°. The theoretical angel *θ_T_* obtained by Equation (5) is 58.6°, 44.4°, 33.9°, and 29.6°. The maximum of conversion efficiency *P*_max_ is 6.06%, 3.45%, 2.22%, and 1.55%, respectively.

Figure 7e shows that the total loss *L_T_* of the double-ribbon bend varying with different bending angles under different *R*: 300 nm (pink line), 400 nm (yellow line), 500 nm (blue line), and 600 nm (green line). *L_T_* increases not only with increasing bending angles under the same *R*, but also increases with increasing *R* under the same bending angle. Because a longer propagating length will be obtained by a larger bending angle under the same *R* and larger *R* under the same bending angle, so does the absorption loss. As can be seen in Figure 7f, the normalized loss *L*_t_ and *L*_b_ both decrease with increasing *R*, but the normalized loss *L*_a_ is nearly unchanged with *R*. Because the *E_f_*, *D*, and *λ* is the same, Im(*n_eff_*) of AEM and SEM is the same, the normalized absorption loss *L*_a_ is accordingly unchanged with *R*. Because a larger *R* has a smaller curvature and lower bending radiation loss, *L*_b_ decreases with increasing *R* and so does *L*_t_. To improve the mode conversion, gain medium can be added to reduce the absorption loss [35].

## 4. Conclusions

In this paper, we propose a new kind of double-ribbon bend that supports two EGSPs and focus on the conversion between two EGSPs. Using FEM, we prove the period of the conversion between two EGSPs is determined by a separation distance, incidence wavelength, and bending radius, and the period of the conversion can be also tuned by the Fermi level. The mode conversion originates from the interference between the two edge plasmonic modes. The loss of double-ribbon bend consists of the absorption loss and bending loss, and the effects of parameters on loss are also discussed. Our double-ribbon bend may provide a new perspective to understand the conversion relationship between two EGSPs. 

## Figures and Tables

**Figure 1 materials-12-04008-f001:**
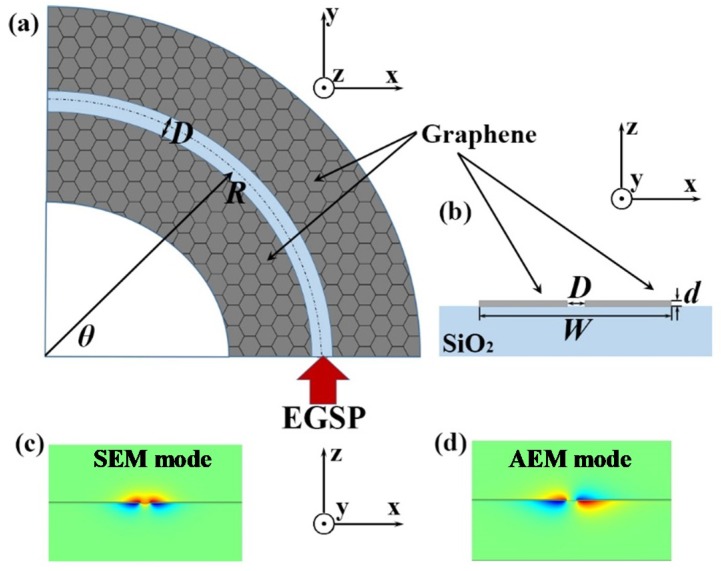
The top-view (*x*-*y* plane) illustration (**a**) and sectional (*x*-*z* plane) profile (**b**) of the designed structure. *Z*-component of the magnetic field distribution for SEM mode in *x*-*z* plane (**c**) and AEM mode (**d**).

**Figure 2 materials-12-04008-f002:**
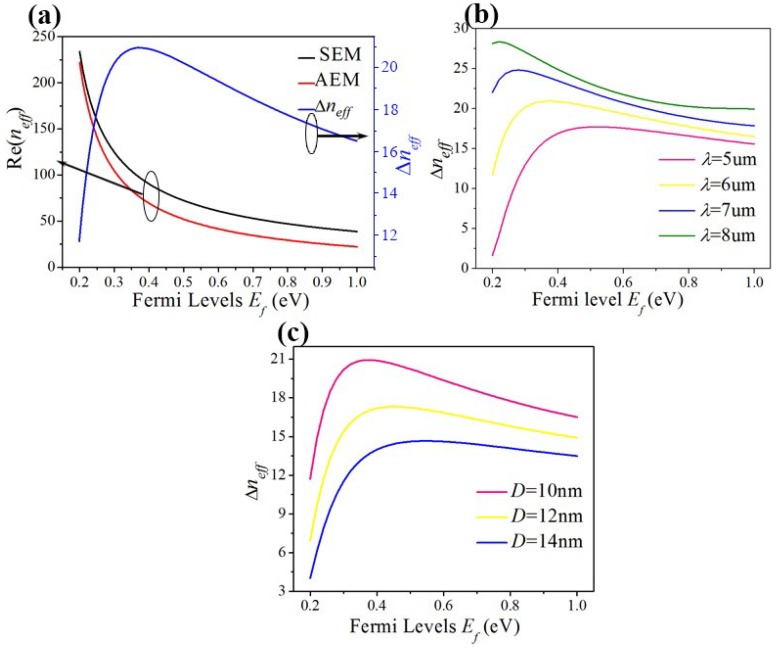
(**a**) the Re(*n_eff_*) of SEM (black line) and AEM (red line), the difference (Δ*n_eff_*) (blue line) between SEM and AEM vs. different *E_f_*; (**b**) Δ*n_eff_* vs. *E_f_* under different incident wavelength *λ*; (**c**) Δ*n_eff_* vs. *E_f_* under different *D*.

**Figure 3 materials-12-04008-f003:**
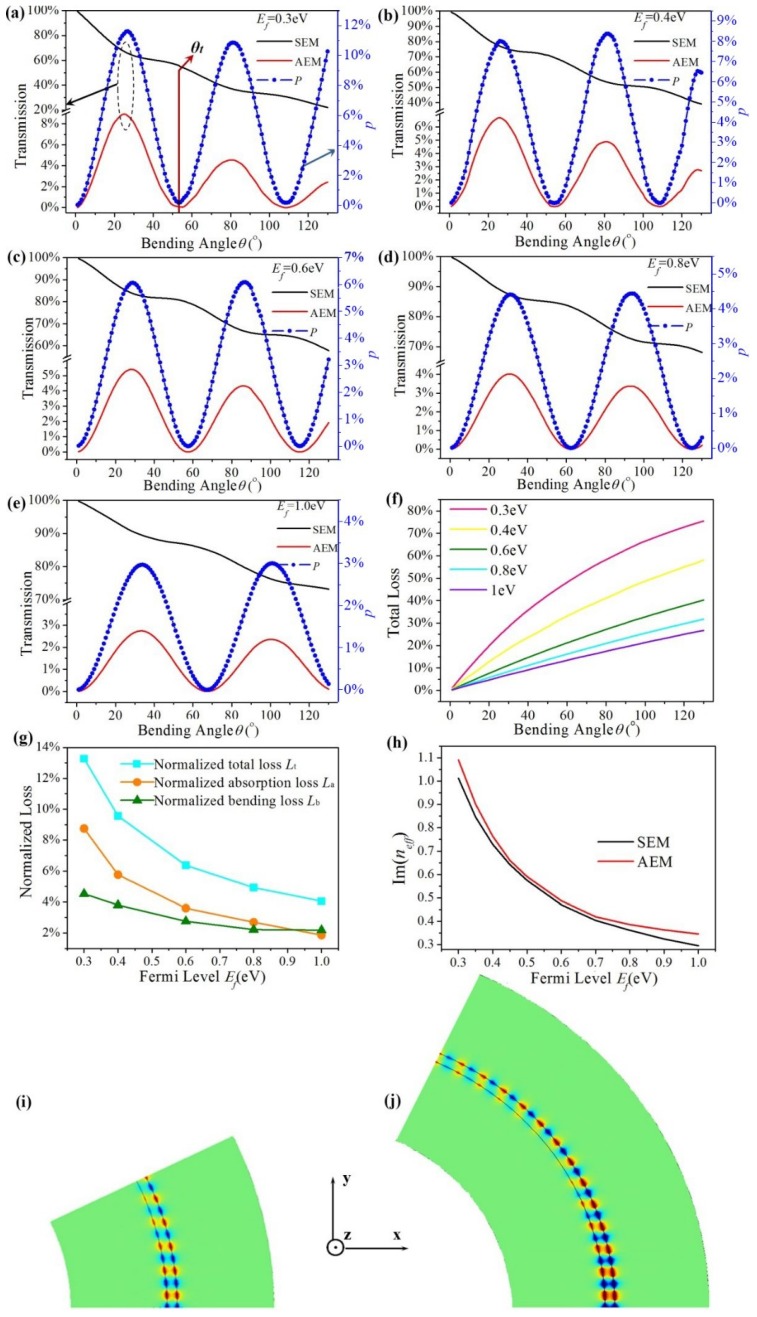
(**a**–**e**) The transmission of the graphene ribbon bends vs. bending; (**f**) total loss of proposed structure vs. bending angle; (**g**) the normalized loss (*L*_t_, *L*_a_ and *L*_b_) under different Fermi levels; (**h**) Im(*n_eff_*) of SEM and AEM varying with *E_f_*; (**i**) and (**j**) the magnetic field distributions *H*_z_ of *x-y* plane when bending angle is 27.3° and 54.5°.

**Figure 4 materials-12-04008-f004:**
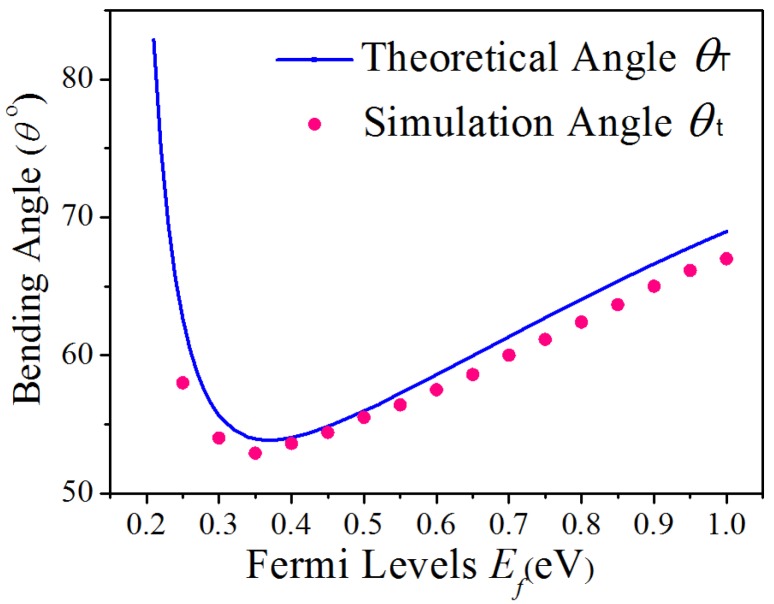
The theoretical period angle *θ_T_* (solid line) and numerical simulation angle *θ_t_* (magenta point) under different Fermi levels.

**Figure 5 materials-12-04008-f005:**
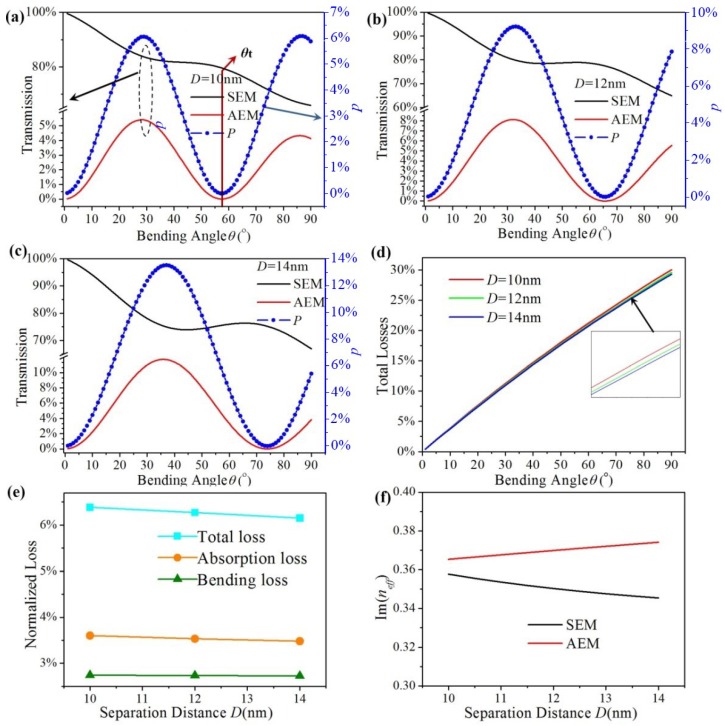
(**a**–**c**) The transmission and conversion efficiency *P* of the graphene ribbon bends vs. bending angle under different *D*. (**d**) total loss of proposed structure vs. bending angle under different *D*. (**e**) the normalized loss (*L*_t_, *L*_a_ and *L*_b_) under three different *D*. (**f**) Im(*n_eff_*) of SEM and AEM varying with *D*.

**Figure 6 materials-12-04008-f006:**
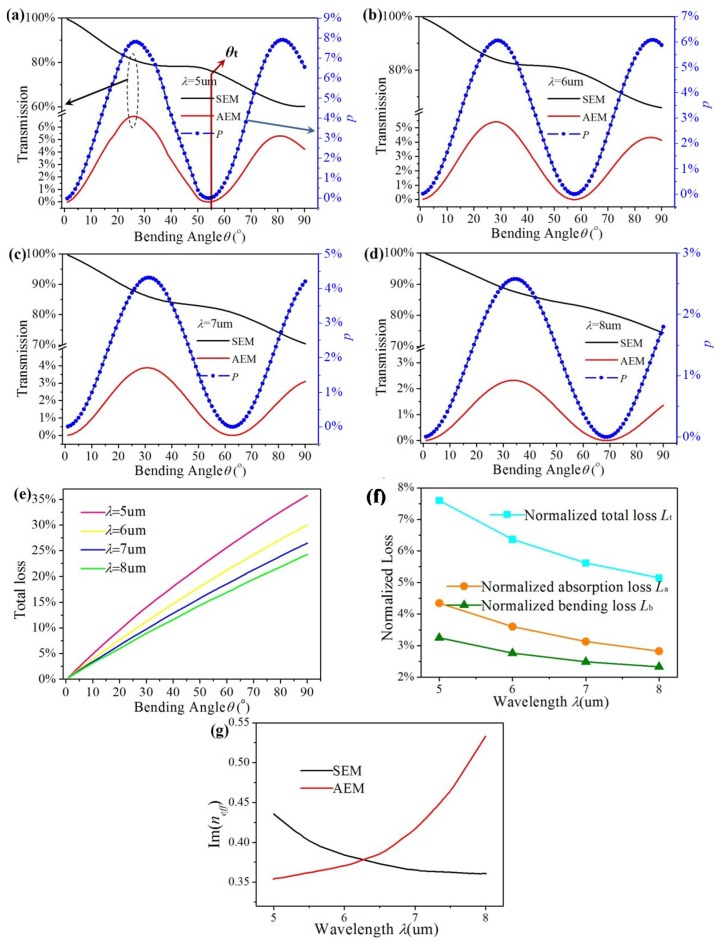
(**a**–**d**) The transmission and conversion efficiency of the graphene ribbon bends vs. bending angle under; (**e**) total loss of proposed structure vs. bending angle; (**f**) the normalized loss (*L*_t_, *L*_a_ and *L*_b_) under different wavelengths. (**g**) Im(*n_eff_*) of SEM and AEM varying with *λ*.

**Figure 7 materials-12-04008-f007:**
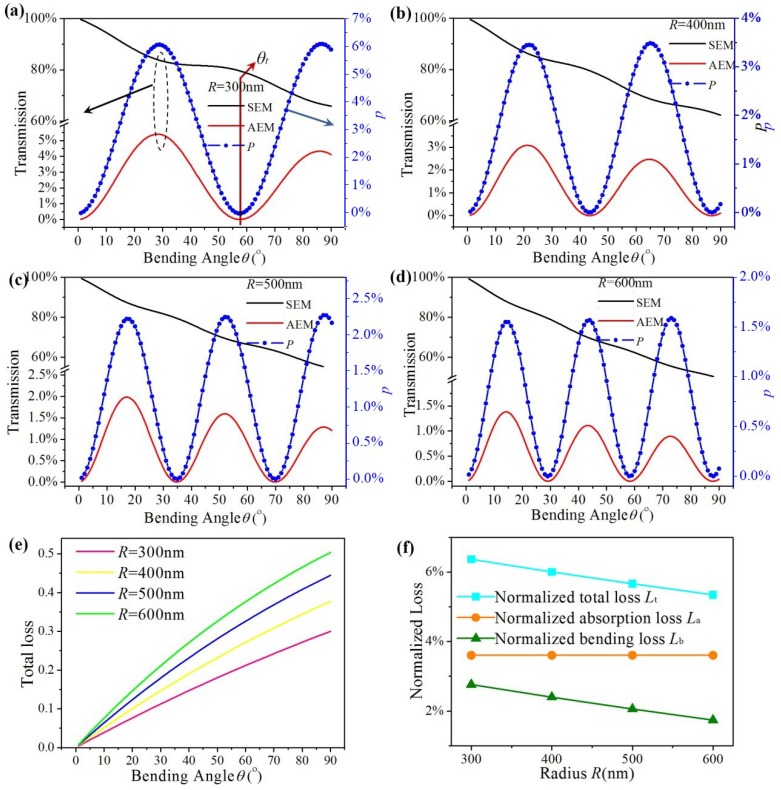
(**a**–**d**) The transmission and conversion efficiency of the graphene ribbon bends vs. bending angle; (**e**) total loss of proposed structure vs. bending angle; (**f**) the normalized loss (*L*_t_, *L*_a_ and *L*_b_) under different *R*.

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
