# Peer review of "Mode Conversion of the Edge Modes in the Graphene Double-Ribbon Bend"

_materials, 2019, doi:10.3390/ma12234008_

Round 1

Reviewer 1 Report

In this manuscript, the authors propose a new types of graphene double-ribbon bend structure that can support surface plasmons mode and investigate the effect of bending angle on the mode conversation at different wavelength and bending radius and investigate optical properties in the mid-IR ranges. Minor points to consider as:

Define EGSPs in the introduction.

It’s fine to briefly re-state/rephrase the text from the abstract section to the conclusion section. However, significant portions of the abstract are overlapped with the introduction section (last paragraph of the introduction) and to the conclusion section, and just rephrased.

The conclusion section should include only the main summary of the results. Authors may wish to increase the font size of the labels in the figure, cite equations if they were borrowed from another literature, label curves and peaks where necessary (please use consistent symbol (either line or dots) to differentiate theoretical result with the simulated ones, see, Fig. 4).

Redundancy: The information given in the text has been repeated in the Figure captions. authors should provide short and clear information in the captions, if the figure has been already described in detail in the text.

Author Response

Thank you very much for your supervision of the reviewing process of my manuscript named “Mode Conversion of the Edge Modes in the Graphene Double-Ribbon Bend” (Manuscript ID materials-621965). We also highly appreciate the reviewer’s carefulness, conscientious, and the broad knowledge on the relevant research fields, since they have given me some beneficial suggestions. According to the reviewer’s instructions, we have made the following revisions on this manuscript. We correct the manuscript point to point, and resubmit my paper. All the corrections are highlighted in red.

Reviewer 1:

In this manuscript, the authors propose a new types of graphene double-ribbon bend structure that can support surface plasmons mode and investigate the effect of bending angle on the mode conversation at different wavelength and bending radius and investigate optical properties in the mid-IR ranges. Minor points to consider as:

Define EGSPs in the introduction.

Reply to: As suggested, we added “EGSPs are the fundamental modes, which are strongly localized along graphene edge, show larger effective refractive index and stronger field confinement 23)” to define EGSPs in the introduction.

It’s fine to briefly re-state/rephrase the text from the abstract section to the conclusion section. However, significant portions of the abstract are overlapped with the introduction section (last paragraph of the introduction) and to the conclusion section, and just rephrased.

Reply to: As suggested, we rewrote the abstract, conclusion, and the last paragraph of the introduction

The conclusion section should include only the main summary of the results. Authors may wish to increase the font size of the labels in the figure, cite equations if they were borrowed from another literature, label curves and peaks where necessary (please use consistent symbol (either line or dots) to differentiate theoretical result with the simulated ones, see, Fig. 4).

Reply to: As suggested, we rewrote the conclusion, which includes only the main summary of results. We also increase the size of labels, and the equation (1) of the surface conductivity of graphene is borrowed from reference 32. The theoretical result in only shown in Fig. 4, and there is no theoretical result in other figures.

Redundancy: The information given in the text has been repeated in the Figure captions. authors should provide short and clear information in the captions, if the figure has been already described in detail in the text.

Reply to: As suggested, we rewrote the captions with short and clear information

Reviewer 2 Report

 In this work the authors used the finite element method to calculate the transmission and loss of edge graphene surface plasmons propagating along the graphene double-ribbon bend in the mid-infrared range. The work is interesting but not sufficiently clear and can be considered for publication if the following revisions are considered:

The abstract should be revised. Definitions as in “different bending angle θ under different Fermi level Ef, separation distance D, incidence wavelength λ, bending radius R and double-ribbon width W.” should not be presented in the abstract. The abstract should also be included the interested of the work developed. Schema of Figure 1 should be better described in the section 2 or the schema should be improved. In line 61, the authors state “As illustrated in Figure 1(a), a pair of paralleled graphene ribbon bends with a separation interval of D is deposited on the SiO2 substrate, and these two bending ribbon are of equal width.”, but figure 1 (a) shows to circular sheets of graphene. In line 88, the authors wrote: “We choose COMSOL Multiphysics based on the finite element method (FEM) to conduct modes analysis of GESPs mode for our double-ribbon bend and all simulation in the paper will be conducted by FEM.” Please give more information and add references. The method of calculation should be better explained. In line 91, the authors wrote:” Z-component of magnetic field in Fig. 1(c) is symmetrical, which can illustrate the symmetric EGSPs mode (SEM); and that of Fig.1 (d) is anti-symmetrical, which can illustrate the anti-symmetric EGSPs mode (AEM).” Please explain the meaning of these of SEM and AEM modes and add references if necessary. In the conclusions section, please do not present definitions as “Fermi level Ef, separation distance D, incidence wavelength λ, and bending radius R”, here the principal conclusions should be presented.

Author Response

Thank you very much for your supervision of the reviewing process of my manuscript named “Mode Conversion of the Edge Modes in the Graphene Double-Ribbon Bend” (Manuscript ID materials-621965). We also highly appreciate the reviewer’s carefulness, conscientious, and the broad knowledge on the relevant research fields, since they have given me some beneficial suggestions. According to the reviewer’s instructions, we have made the following revisions on this manuscript. We correct the manuscript point to point, and resubmit my paper. All the corrections are highlighted in red.

Reviewer 2:

In this work the authors used the finite element method to calculate the transmission and loss of edge graphene surface plasmons propagating along the graphene double-ribbon bend in the mid-infrared range. The work is interesting but not sufficiently clear and can be considered for publication if the following revisions are considered:

The abstract should be revised. Definitions as in “different bending angle θ under different Fermi level Ef, separation distance D, incidence wavelength λ, bending radius R and double-ribbon width W.” should not be presented in the abstract. The abstract should also be included the interested of the work developed.

Reply to: As suggested, we deleted the definitions in the abstract.

Schema of Figure 1 should be better described in the section 2 or the schema should be improved. In line 61, the authors state “As illustrated in Figure 1(a), a pair of paralleled graphene ribbon bends with a separation interval of D is deposited on the SiO2 substrate, and these two bending ribbon are of equal width.”, but figure 1 (a) shows to circular sheets of graphene.

Reply to: Figure 1(a) shows a pair of graphene ribbon bends, and the width of one ribbon is 95nm ((W-D)/2), shown as in Fig. 1(b). And we improve the description of Fig. 1(a), in the revised manuscript.

In line 88, the authors wrote: “We choose COMSOL Multiphysics based on the finite element method (FEM) to conduct modes analysis of GESPs mode for our double-ribbon bend and all simulation in the paper will be conducted by FEM.” Please give more information and add references.

Reply to: For giving more information, we add reference 28 in the revised manuscript. We use the method of reference 28, which is our previous work, to conduct the modes analysis of GESPs mode and all simulation.

The method of calculation should be better explained. In line 91, the authors wrote:” Z-component of magnetic field in Fig. 1(c) is symmetrical, which can illustrate the symmetric EGSPs mode (SEM); and that of Fig.1 (d) is anti-symmetrical, which can illustrate the anti-symmetric EGSPs mode (AEM).” Please explain the meaning of these of SEM and AEM modes and add references if necessary.

Reply to: The edge mode, whose Z-component of magnetic field is symmetrical, is defined as symmetric EGSPs mode (SEM); the edge mode, whose Z-component of magnetic field is anti-symmetrical, is defined as symmetric EGSPs mode (SEM). This definition is similar with Ref. 31.

In the conclusions section, please do not present definitions as “Fermi level Ef, separation distance D, incidence wavelength λ, and bending radius R”, here the principal conclusions should be presented.

Reply to: For giving more information, we delete the definitions in the conclusions section, and rewrote the conclusion.

Reviewer 3 Report

Line 25: dash between the and art.

Figure 2 presents the graphs more clearly. Figure 2a has two different y-axes represent it with a different color. Fonts are not visible. Put the (a). (c) inside the figure.

Figures having two Y use a different color so that easily identifiable.

For equation (7), define R1 and R2. Mention of T1 and T2 should be there after the equation.

“Fig. 3(g) shows the imaginary parts of the effective” Is it 3(g)!

Conclusion: Include more. Too short, elaborate.

Author Response

Thank you very much for your supervision of the reviewing process of my manuscript named “Mode Conversion of the Edge Modes in the Graphene Double-Ribbon Bend” (Manuscript ID materials-621965). We also highly appreciate the reviewer’s carefulness, conscientious, and the broad knowledge on the relevant research fields, since they have given me some beneficial suggestions. According to the reviewer’s instructions, we have made the following revisions on this manuscript. We correct the manuscript point to point, and resubmit my paper. All the corrections are highlighted in red.

Line 25: dash between the and art.

Reply to: we have revised it.

Figure 2 presents the graphs more clearly. Figure 2a has two different y-axes represent it with a different color. Fonts are not visible. Put the (a). (c) inside the figure. Figures having two Y use a different color so that easily identifiable.

Reply to: we have revised all the figures of two y-axes with different colors, and presents the graphs more clearly

For equation (7), define R1 and R2. Mention of T1 and T2 should be there after the equation.

Reply to: T1 and T2 have been defined after equation (6).

“Fig. 3(g) shows the imaginary parts of the effective” Is it 3(g)!

Reply to: we have revised this mistake, and it should be Fig. 3(h).

Conclusion: Include more. Too short, elaborate.

Reply to: we have rewritten the inclusion as suggested, and make the conclusion include more.

Reviewer 4 Report

This paper investigated mode conversion of the edge modes in the graphene double ribbon bend while various structure parameters, bending angle, Fermi level, separation distance, and incidence wavelengths, etc., are swept. But there are no physical explanation of the optical properties, transmission, absorptions, on the structure parameters. In addition, the mode conversion ratio is only 5-15%, therefore I'm not sure that this mode conversion with low efficiency is useful to design graphene surface plasmon based devices. I recommend this paper should be revised extensively for the publication.

As I mentioned, there are lots of graphs of the transmission, conversion efficiencies, etc., which indicate dependencies on the various structure parameters. However, there are a little physical explanations on the reason of the dependencies. Authors should implement detailed discussions on the reasons of the dependencies. In this paper, the double ribbon design seems extremely difficult to be fabricated. 10 nm-gap size should be maintained in the bend region. Is it reasonable design to fabricate? Are only 5-15% conversion efficiencies enough for the proposed design to be used in the Photonic integrated circuit based on graphene? Please explain the meaning of the conversion efficiency. Since many structure parameters are considered to calculate the conversion efficiency, what can be the optimized design for the highest conversion efficiency?

Author Response

Thank you very much for your supervision of the reviewing process of my manuscript named “Mode Conversion of the Edge Modes in the Graphene Double-Ribbon Bend” (Manuscript ID materials-621965). We also highly appreciate the reviewer’s carefulness, conscientious, and the broad knowledge on the relevant research fields, since they have given me some beneficial suggestions. According to the reviewer’s instructions, we have made the following revisions on this manuscript. We correct the manuscript point to point, and resubmit my paper. All the corrections are highlighted in red.

As I mentioned, there are lots of graphs of the transmission, conversion efficiencies, etc., which indicate dependencies on the various structure parameters. However, there are a little physical explanations on the reason of the dependencies. Authors should implement detailed discussions on the reasons of the dependencies. In this paper, the double ribbon design seems extremely difficult to be fabricated. 10 nm-gap size should be maintained in the bend region. Is it reasonable design to fabricate? Are only 5-15% conversion efficiencies enough for the proposed design to be used in the Photonic integrated circuit based on graphene? Please explain the meaning of the conversion efficiency. Since many structure parameters are considered to calculate the conversion efficiency, what can be the optimized design for the highest conversion efficiency?

Reply to: Thanks a lot for your beneficial suggestion. I believe that we have given the physical explanation of the optical properties, transmission, absorptions, on the structure parameters. The mode conversion originates from the interference between the two EGSPs, and the loss mainly originates from the of EGSPs. The fabricate process can be achieved by the following steps: a high-quality large-area graphene film can be obtained using an optimized liquid precursor chemical vapor deposition method [1]. The film is transferred onto a silica substrate, and the double-ribbon bend pattern can be fabricated by electron beam lithography and oxygen plasma etching [2], although it is a difficult process. The 5-15% conversion efficiencies are low for photonic integrated circuit, but here we focus on the physics and phenomenon, and we believe that the mode conversion of the double-ribbon bend is firstly discussed by this paper. Improving the conversion efficiencies are also our following work. Besides, the conversion efficiency has been defined by Eq. (6)

Round 2

Reviewer 2 Report

 The authors have done the suggested revision therefore the article now can be considered for publication.

Author Response

As suggested, we have checked the English language and style, and revised the  spell mistakes.

Reviewer 4 Report

Authors revised the paper carefully by commenting all the issues raised by the reviewer. Most of the comments of the reviewer were properly answered. However, only one thing should be included for the publication. In the paper, the best design for the high mode conversion and the method to increase the conversion efficiency should be commented briefly.

Author Response

As suggested, we add"To improve the mode conversion, gain medium can be added to reduce the absorption loss35)." in the line 236 to propose a method to increase the conversion efficiency. And we add the reference 35.